# A Genetic Trap in Yeast for Inhibitors of SARS-CoV-2 Main Protease

Hanna Alalam,[a] Sunniva Sigurdardóttir,[a] Catarina Bourgard,[a] Ievgeniia Tiukova,[b] Ross D. King,[b] Morten Grøtli,[a] Per Sunnerhagen[a]

[a]Department of Chemistry and Molecular Biology, University of Gothenburg, Göteborg, Sweden
[b]Department of Biology and Biological Engineering, Chalmers, Göteborg, Sweden

**ABSTRACT** The ongoing COVID-19 pandemic urges searches for antiviral agents that can block infection or ameliorate its symptoms. Using dissimilar search strategies for new antivirals will improve our overall chances of finding effective treatments. Here, we have established an experimental platform for screening of small molecule inhibitors of the SARS-CoV-2 main protease in *Saccharomyces cerevisiae* cells, genetically engineered to enhance cellular uptake of small molecules in the environment. The system consists of a fusion of the *Escherichia coli* toxin MazF and its antitoxin MazE, with insertion of a protease cleavage site in the linker peptide connecting the MazE and MazF moieties. Expression of the viral protease confers cleavage of the MazEF fusion, releasing the MazF toxin from its antitoxin, resulting in growth inhibition. In the presence of a small molecule inhibiting the protease, cleavage is blocked and the MazF toxin remains inhibited, promoting growth. The system thus allows positive selection for inhibitors. The engineered yeast strain is tagged with a fluorescent marker protein, allowing precise monitoring of its growth in the presence or absence of inhibitor. We detect an established main protease inhibitor by a robust growth increase, discernible down to 1 $\mu$M. The system is suitable for robotized large-scale screens. It allows *in vivo* evaluation of drug candidates and is rapidly adaptable for new variants of the protease with deviant site specificities.

**IMPORTANCE** The COVID-19 pandemic may continue for several years before vaccination campaigns can put an end to it globally. Thus, the need for discovery of new antiviral drug candidates will remain. We have engineered a system in yeast cells for the detection of small molecule inhibitors of one attractive drug target of SARS-CoV-2, its main protease, which is required for viral replication. The ability to detect inhibitors in live cells brings the advantage that only compounds capable of entering the cell and remain stable there will score in the system. Moreover, because of its design in yeast cells, the system is rapidly adaptable for tuning the detection level and eventual modification of the protease cleavage site in the case of future mutant variants of the SARS-CoV-2 main protease or even for other proteases.

**KEYWORDS** COVID-19, small molecules, *Saccharomyces cerevisiae*, genetic engineering, MazF toxin, SARS-CoV-2, antiviral agents, genetic selection system

The COVID-19 pandemic caused by severe acute respiratory syndrome coronavirus 2 (SARS-CoV-2) has been underway for a year and a half. Despite unparalleled successes in vaccine development and the rollout of vaccination programs, it is uncertain when these efforts will be sufficient to quell the outbreak in the face of arising mutant virus strains. For the foreseeable future, the need for antiviral drugs and treatment for COVID-19 remains. The initial attempts at drug discovery have been dominated by repurposing antiviral and other drugs. Previous experience of discovery and the development of antiviral drugs against coronaviruses is limited. The SARS epidemic in 2003,

Address correspondence to Per Sunnerhagen, per.sunnerhagen@cmb.gu.se.

as well as the outbreak of Middle East respiratory syndrome (MERS) in 2012, did spur some early drug discovery efforts (1–3). However, none of those efforts have been further developed into clinically approved antiviral drugs. Given the similarity between the coronaviruses MERS-CoV, SARS-CoV-1, and SARS-CoV-2, it is likely that an antiviral drug effective against any of the first two could also be effective against SARS-CoV-2. This argues that continued research through multiple avenues on new antiviral drug candidates targeting SARS-CoV-2 will be warranted for the next several years.

The coronavirus main protease (Mpro; also known as poliovirus 3C-like protease [3CL pro]) cleaves the viral polyprotein at 11 sites into its individual functional protein products and is essential for the virus to replicate. It is recognized as a suitable target by virtue of its critical role in viral propagation, and by the well-explored druggability of proteases. There are no human host cell proteases structurally related to SARS-CoV-2 Mpro, and no human protease shares its strong preference for cleaving C-terminally of a glutamine. In contrast, the SARS-CoV-2 papain-like protease—also an antiviral target candidate—is expected to have overlapping substrate and inhibitor sensitivity profile with host cell deubiquitinases (4).

The crystal structure of Mpro has been determined (5), opening up for structure-based drug discovery. However, it has been argued that the dynamic properties of SARS-CoV-2 Mpro, in particular through flexible regions near the catalytic active site, make it a more difficult target for rational drug discovery than the SARS-CoV-1 Mpro (6).

An *in vivo* assay of candidate inhibitors designed to act intracellularly confers the advantage that only molecules able to be efficiently taken up by cells and remain stable in the intracellular environment will score in the assay. Testing of candidate molecules in assays with purified enzyme *in vitro* is sensitive to buffer conditions, such as recently demonstrated for putative SARS-CoV-2 Mpro inhibitors tested in a variety of *in vitro* assays, where the apparent inhibition was obliterated by the restoration of a reducing environment (7). Using yeast (*Saccharomyces cerevisiae*) gives access to a versatile platform for precise and rapid genetic engineering of intracellular testing systems. Genetic screens in yeast have previously identified small molecule inhibitors to, e.g., HIV-1 protease (8), SARS-CoV-1 papain-like protease (9), and SARS-CoV-1 mRNA cap-methyltransferase (10).

Here, we have constructed a yeast strain expressing a synthetic fusion protein, which upon cleavage by a protease releases a bacterial toxin moiety capable of inhibiting growth of the yeast strain. In the same strain, SARS-CoV-2 Mpro is expressed, and its recognition site is inserted in the fusion protein, making cell growth negatively regulated by the Mpro activity level, but stimulated by Mpro inhibitors (Fig. 1). The yeast strain genetic background is sensitized to external molecules by inactivation of membrane-bound transporters. We demonstrate detection of an established Mpro inhibitor by growth promotion of engineered yeast cells, in concentrations down to 1 $\mu$M. The system is suited for physical screening in robotized setups that allow controlled growth of yeast cells in microtiter format and measurement of fluorescence intensity. The design enables rapid upgrades of the protease recognition site according to eventual future mutant variants of the virus.

## RESULTS

**Construction of a genetic trap in yeast.** The *E. coli* toxin MazF is a single-strand endoribonuclease that cleaves mRNA at 5'-ACA-3' sequences (11). When expressed in either a prokaryotic or eukaryotic cell, it arrests growth or kills that cell, depending on the expression level. MazF is a component of a chromosomally encoded toxin/antitoxin system in *E. coli*, the other component being MazE, a protein binding to and inhibiting MazF. MazE has a much shorter turnover time than MazF, requiring the cell to continuously synthesize MazE to maintain viability (12).

It has previously been demonstrated that a synthetic protein fusion of a C-terminal fragment of MazE to MazF resulted in a chimeric protein with inhibition of MazF activity, permitting cell survival and growth also at high expression levels (13). Insertion of a protease site in the added linker sequence connecting the two protein moieties permit

mSystems®

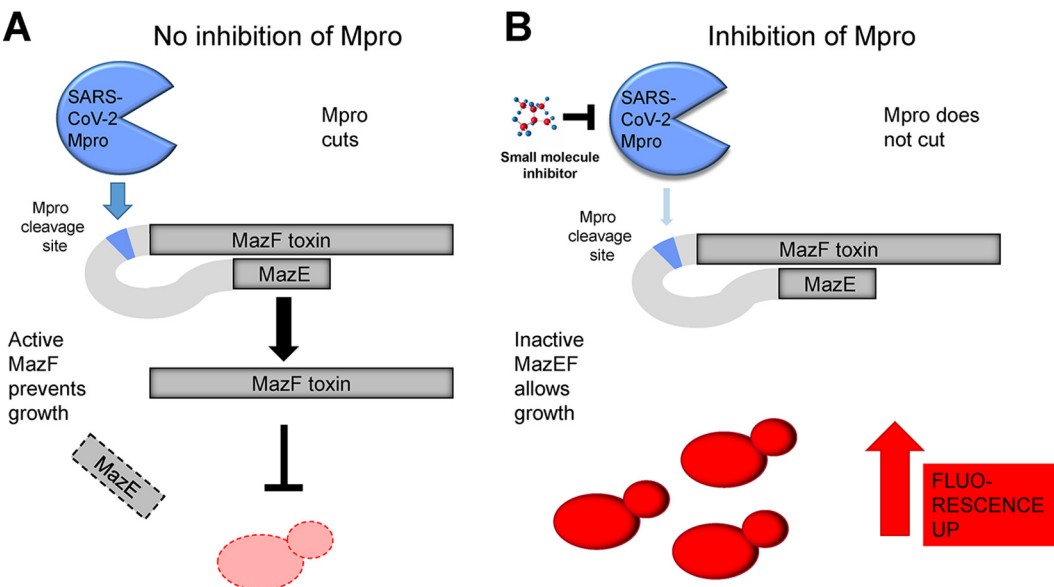

**FIG 1** Principle of the genetic selection system. (A) Situation without Mpro inhibitor. SARS-CoV-2 Mpro is active and cuts the inactive MazEF fusion protein at the synthetic Mpro cleavage site in the linker region connecting the MazF (toxin) and MazE (toxin inhibitor) moieties, releasing active MazF. The RNase activity of MazF can then exert its toxic effect by degrading cellular RNAs, thus preventing growth of the cells carrying the construct. The output signal from the red fluorescent protein tag in these cells will be weak. (B) Situation with Mpro inhibitor. The activity of SARS-CoV-Mpro is reduced, and the inactive MazEF fusion stays mostly intact. The toxic RNase activity will be reduced, and these cells can grow faster, resulting in a stronger fluorescent signal output.

release of fully active MazF upon expression of the protease and cleavage of the linker peptide (13). To exploit these findings for creation of a genetic selection system in *S. cerevisiae*, we generated a chimeric construct (MazEF) with the Mpro cleavage site in the linker. Specifically, the construct contained the 41 C-terminal amino acid residues Leu42 to Trp81 of MazE with an added ATG codon preceding the Leu42 codon, followed by a GGVKLQSGS amino acid linker sequence containing the Mpro cleavage site VKLQS, joint N-terminally to the full amino acid sequence of MazF. The entire coding region was modified using silent mutations, making it devoid of ACA sequences to reduce cleavage of *MazF* mRNA. The coding sequence of this fusion was put under transcriptional control of the *MET3* promoter in a pCM188 (14) backbone or under the control of the *GAL1* promoter in the same backbone in order to achieve two ranges of expression, weak and strong, respectively (Table 1).

**TABLE 1** Strains and plasmids used in this study

| Strain or plasmid | Relevant genotype/selection | Comment | Source or reference |
|---|---|---|---|
| Strains | | | |
| 1352-Y13363 | *MATα his3Δ1 leu2Δ0 met15Δ0 ura3Δ0 pdr1:: NatMX pdr3::Kl.URA3 snq2Δ::Kl.LEU2* | Parent strain for construction | Cleslei Zanelli |
| HA_SC_1352control | *1352-Y13363 pdr3::MET17* | Control strain without Mpro | This study |
| HA_SC_Met17_Mpro | *1352-Y13363 pdr3::MET17+Mpro* | Strain expressing Mpro | This study |
| Plasmids | | | |
| PSMv3-GAL | *URA3* | Toxin-antitoxin fusion driven by *GAL1* promoter | This study |
| pCM188-MET3 | *URA3* | Empty vector, *MET3* promoter | This study |
| PSMv4 | *URA3* | Toxin-antitoxin fusion driven by *MET3* promoter | This study |
| YEP_CHERRY_HIS3 | *HIS3* | Plasmid expressing mCherry | 23 |
| HA_SC_1352control_RED | *1352-Y13363 pdr3::MET17 his3Δ1:: HIS3+mCHERRY* | Control strain expressing mCherry | This study |
| HA_SC_Met17_Mpro_RED | *1352-Y13363 pdr3::MET17+Mpro his3Δ1:: HIS3+mCHERRY* | Strain expressing Mpro and mCherry | This study |

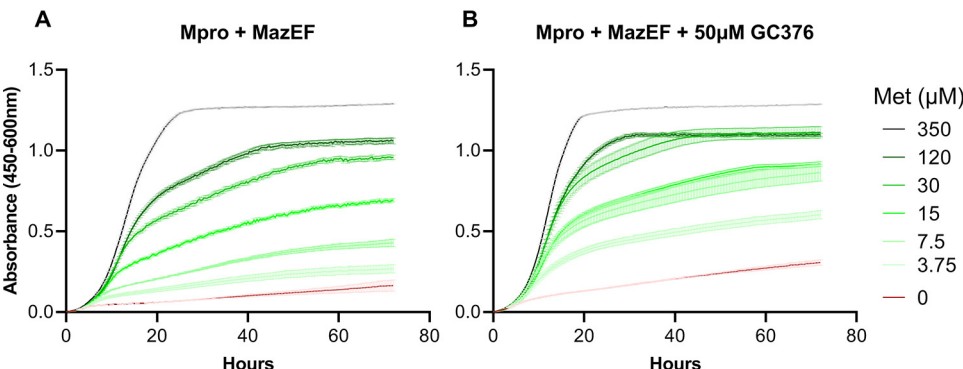

**FIG 2** Titration of methionine concentration with a yeast strain expressing Mpro and MazEF chimera (strain HA_SC_Met17_Mpro carrying PSMv4) in the presence or absence of GC376. Growth measurements were obtained using the Bioscreen C reader, with the starting absorbance adjusted to 0 (mean ± SE, n = 3). (A) Cell cultures were grown in SD medium without uracil (SD–ura) and with various concentrations of methionine from 350 μM to 0 μM (see the legend). (B) Same conditions as for panel A but treated with the protease inhibitor GC376 (50 μM).

The genetic background for strain constructions was *S. cerevisiae* strain 1352-Y13363 (Table 1). It carries null alleles of *PDR1* and *PDR3*, encoding transcriptional regulators of a large number of ABC transporters exporting small molecules from the cell, and of *SNQ2*, which encodes an ABC exporter protein important for detoxification (15). This triple mutation greatly increases the number of compounds to which yeast cells are sensitive, without compromising vigor (16). In this strain, the *Kluyveromyces lactis URA3* marker disrupting the *PDR3* locus was first replaced by *S. cerevisiae MET17* to create the control strain and allow methionine repressible expression from the *MET3* promoter (17) by restoring methionine prototrophy. The Mpro-expressing strain was constructed by ligating a synthetic fragment carrying Mpro under the control of the *Pichia GAP* promoter with *MET17* and integrating in the *PDR3* locus (see Text S1 in the supplemental material). The strains were tagged with the red-fluorescent mCherry marker under the control of the constitutive *TDH3* promoter, which was integrated into the *HIS3* locus. Full strain construction details are given in Text S1.

**Responsiveness of engineered strains to the GC376 protease inhibitor.** We wanted to evaluate the engineered yeast strains for their usefulness as a testing platform for potential Mpro inhibitors. To do this, we tested the impact of expressing MazEF at different levels in the presence of Mpro and of externally added small molecules.

GC376 is a broad-spectrum antiviral molecule active against 3CL-like proteases from coronaviruses (18) which has been used to treat feline infectious peritonitis (19). GC376 is a prodrug and was recently shown to be effective against SARS-CoV-2 Mpro *in vitro* (20, 21) and against SARS-CoV-2 in human cell culture (21, 22). We chose this compound as a positive control for an Mpro inhibitor to titrate the growth response.

We first investigated a strain expressing the MazEF chimera from the *GAL1* promoter and Mpro from the *Pichia GAP* promoter (see Text S1). However, the addition of CG376 even at 100 μM only marginally improved growth in inducing galactose medium (see Fig. S1). We concluded that expression of the MazEF toxin fusion protein from the strong *GAL1* promoter in the presence of Mpro was too detrimental to be overcome by protease inhibitors. Thus, this expression system is impractical for screening purposes.

Reasoning that we needed weaker and regulatable expression of the toxin chimera, we next examined the behavior of strains expressing Mpro from the *Pichia GAP* promoter and the toxin chimera from the *MET3* promoter in various methionine concentrations. As seen in Fig. 2A, growth of the strain expressing both Mpro and MazEF is increasingly depressed by lower methionine concentrations in the range below 350 μM. This is expected since *MET3* is negatively regulated by methionine (17). In

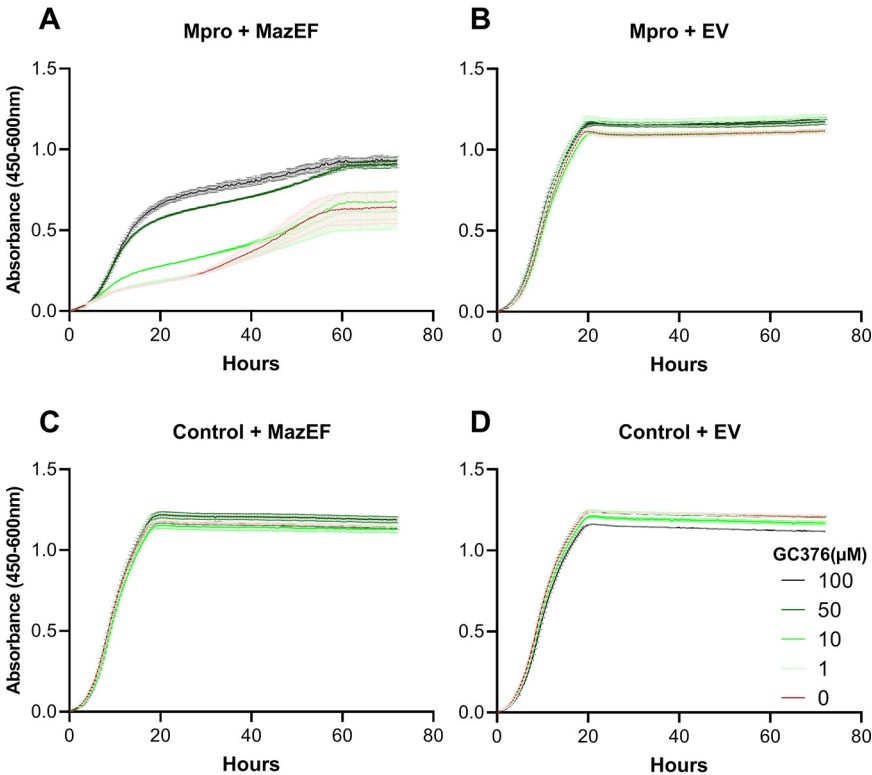

**FIG 3** Titration of inhibitor GC376. Growth measurements were obtained as in Fig. 2 (mean ± SE, *n* = 3). Cell cultures were grown in SD–ura and 7.5 $\mu$M methionine and various concentrations of GC376 (see legend). (A) Strain HA_SC_Met17_Mpro carrying PSMv4 expressing Mpro and MazEF chimera. (B) Strain HA_SC_Met17_Mpro expressing Mpro (EV; empty vector). (C) Strain HA_SC_1352control carrying PSMv4 expressing the MazEF chimera only. (D) Control strain HA_SC_1352control with empty vector.

contrast, the growth of strains expressing only Mpro, only MazEF, or neither is not significantly affected by the methionine concentration (see Fig. S2, left column). This indicates that the negative impact of expression of Mpro alone in yeast cells is negligible under these conditions and that MazEF is inactive and not cleaved to release active MazF in the absence of Mpro, in line with the results with MazEF expression from the *GAL1* promoter (see Fig. S1).

To achieve maximal separation between cells exposed to an Mpro inhibitor, we then grew cells expressing MazEF and Mpro in increasing methionine concentrations up to 350 $\mu$M and compared growth with or without GC376 at 50 $\mu$M (Fig. 2B). As expected, if GC376 inhibits Mpro and therefore prevents cleavage of MazEF, releasing toxic MazF, growth was improved by the presence of GC376. The best absolute separation was seen at 7.5 to 15 $\mu$M methionine. To verify that the improved growth was indeed due to decreased Mpro activity, we then exposed cells to a range of GC376 concentrations up to 100 $\mu$M, maintaining a fixed methionine concentration of 7.5 $\mu$M. GC376 at 50 $\mu$M did enhance growth of the strain expressing both Mpro and MazEF (Fig. 2B and 3A), but as expected had no effect on growth of any of the strains lacking Mpro, MazEF, or both (see Fig. S2 and S3). Finally, we wanted to determine whether lower inhibitor concentrations than 50 $\mu$M could also be detected with strains carrying the mCherry fluorescent marker. With methionine at 10 $\mu$M or lower, we could discriminate the effect of GC376 down to 10 $\mu$M (yield ratio = 1.57, standard error [SE] ± 0.02, *n* = 16, *P* < 2.2E–16; see Fig. S4). This also demonstrates that tagging cells with mCherry did not cause a noticeable change in their response to methionine or GC376 concentration (Fig. 3; see also Fig. S3).

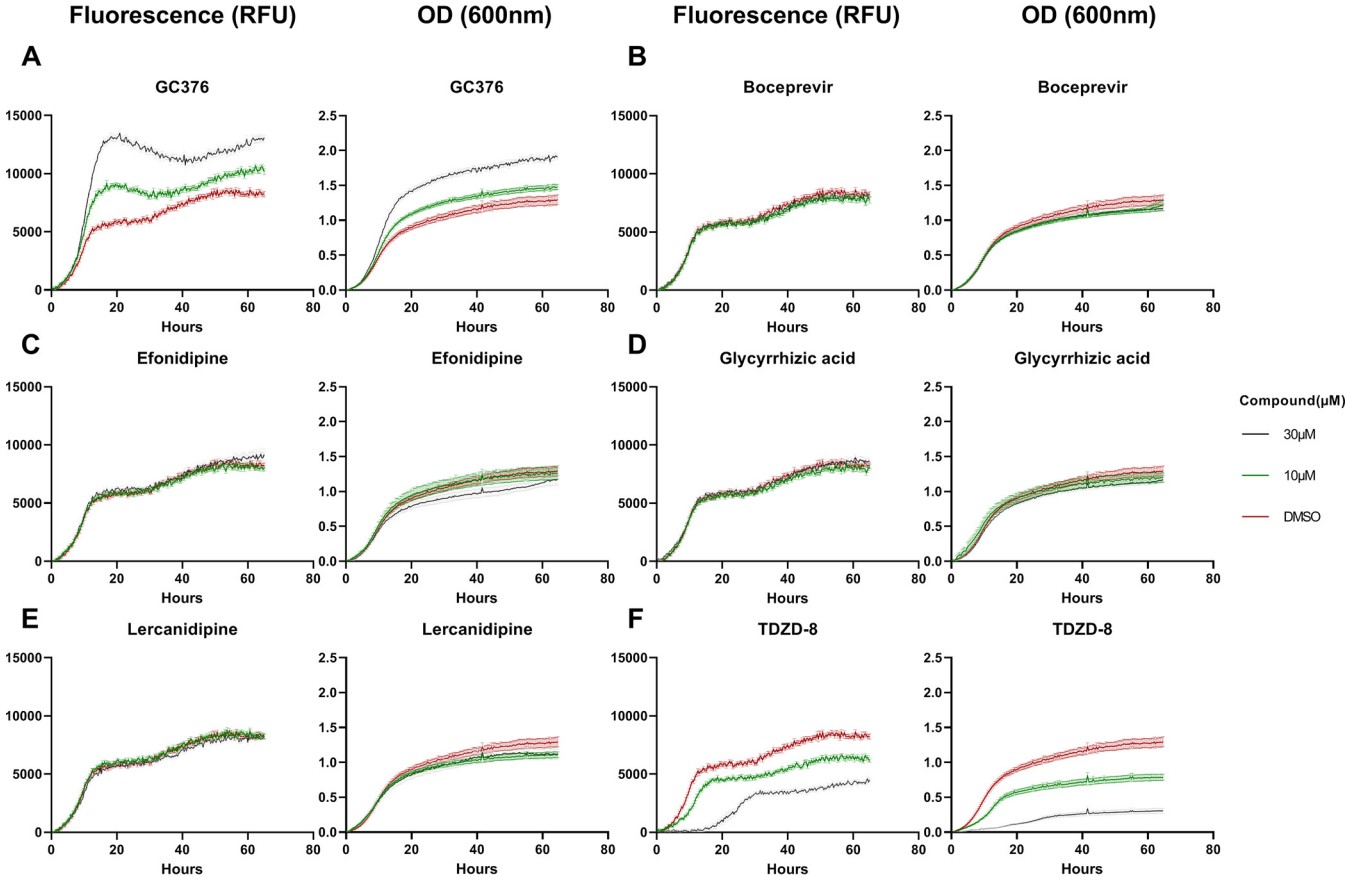

**FIG 4** Growth of mCherry-tagged strain expressing Mpro and MazEF chimera in the presence of different candidate protease inhibitors. Strain HA_SC_Met17_Mpro_RED carrying PSMv4 was used. Growth measurements (fluorescence and absorbance) were obtained using the Eve robot (see Materials and Methods), with the starting measurement adjusted to 0 (mean ± SE, $n = 16$). RFU, relative fluorescence units. Cell cultures were grown in SD–ura and 15 $\mu$M methionine; 1.25% DMSO; and 0, 10, and 30 $\mu$M concentrations of the respective compound to be tested (see the legend).

**Measuring impact of candidate protease inhibitors on strains using a fluorescence readout.** Having tuned the expression levels of the MazEF chimera from the *MET3* promoter to achieve a robust impact on growth by the inhibitor GC376, we wanted to test the screening system in a robotized platform suitable for screening of large compound libraries. Quantifying growth as the fluorescence of a genetically tagged marker protein potentially offers better signal-to-noise ratios than using the optical density (OD). Thus, we monitored the growth of a strain expressing MazEF and Mpro and, in addition tagged with mCherry, in a robotic platform allowing automated growth monitoring of thousands of samples simultaneously and regular fluorescence readings (23).

Besides GC376, we exposed the strain to other compounds implicated by different criteria as potential Mpro inhibitors. Boceprevir is a clinically approved human hepatitis C virus N53 protease inhibitor, showing activity against SARS-CoV-2 Mpro in cultured human cells (22). TDZD-8, originally characterized as an inhibitor of glycogen synthase kinase $\beta$ (GSK3-$\beta$), scored as a SARS-CoV-2 Mpro inhibitor in an *in vitro* assay but was subsequently dismissed as a false positive because of its aggregation properties (24). Efonidipine and lercanidipine are Ca$^{2+}$ channel blockers found as hits in an *in silico* screen for SARS-CoV-2 Mpro inhibitors and also active in an *in vitro* enzymatic assay (25). Glycyrrhizic acid at high concentrations blocks SARS-CoV-2 replication in cell culture and inhibits Mpro in an *in vitro* assay (26). Compounds were applied at 10 and 30 $\mu$M. We adjusted the methionine concentration to 15 $\mu$M to avoid a gradual increase in toxicity caused by methionine consumption toward the end of the run, as can be seen in Fig. 3A at 10 $\mu$M.

In Fig. 4A we see that, based on the fluorescence readings in this setting, the

**TABLE 2** Signal discrimination for detection of the Mpro inhibitor (GC376)[a]

| Concn ($\mu$M) | n | Fluorescence | | OD$_{600}$ | | P[b] |
|---|---|---|---|---|---|---|
| | | Ratio | SE | Ratio | SE | |
| After 20 h | | | | | | |
| 1 | 16 | 1.14 | 0.03 | ND | | 7.0E–3 |
| 10 | 16 | 1.57 | 0.02 | 1.22 | 0.03 | 2.1E–11 |
| 30 | 20 | 2.28 | 0.03 | 1.61 | 0.03 | 2.2E–16 |
| | | | | | | |
| After 65 h | | | | | | |
| 1 | 16 | 1.06 | 0.07 | ND | | 0.5 |
| 10 | 16 | 1.20 | 0.02 | 1.19 | 0.03 | 0.64 |
| 30 | 20 | 1.49 | 0.02 | 1.55 | 0.03 | 0.068 |

[a]Calculated as the ratio of yield compared to controls without inhibitor.
[b]Welsh two-sample t test comparing yield ratios measured by fluorescence.

growth-promoting effect of CG376 on the tester strain is dose dependent and obvious also at 10 $\mu$M. This demonstrates that the system works well also in a setting with limited aeration and fluorescence-based monitoring of growth. For the other compounds, no positive effect is seen at any concentration. TDZD-8 instead displayed a negative impact on growth (Fig. 4F). The rest of the compounds tested—boceprevir, efonidipine, glycyrrhic acid, and lercanidipine—did not impact growth (Fig. 4). The OD readings, which were obtained simultaneously, closely paralleled the fluorescence values. As seen in Table 2, the discrimination for the tester strain between treatment with GC376 and no treatment was best for fluorescence readings at 20 h for both 10 and 30 $\mu$M inhibitor (ratios 1.59 and 2.28, respectively). This is better than what was achieved with the reading OD (ratios 1.22 and 1.61). In order to examine the lower detection limit, we further tested GC376 and boceprevir at 1 and 3 $\mu$M. A discernible growth enhancement around 20 h was observed with GC376 down to 1 $\mu$M (Fig. 5A), which is also statistically significant (Table 2). For boceprevir, no effect at 20 h could be seen even at 200 $\mu$M; however, a moderate stimulation was observed at 100 and 200 $\mu$M toward the end of the experiment (Fig. 5B). This indicates that the previously observed lack of effect at 20 h by boceprevir was not a question of insufficient dosage; however, some minor activity may account for the late effect.

## DISCUSSION

We have created an experimental system for physical screening of candidate inhibitors of SARS-CoV-2 Mpro in yeast cells. We demonstrate that it can be used to detect with good confidence a known Mpro inhibitor at concentrations down to 1 $\mu$M. This sensitivity compares favorably or equally to several systems based on mammalian cells (7, 18, 20, 21). The growth of control strains lacking MazEF, Mpro, or both is unaffected by the inhibitor, showing the specificity of the assay. The system is suitable for screening of chemical libraries in a robot platform capable of cultivating large numbers of yeast cells in controlled conditions with regular monitoring of fluorescence. Using a fluorescent readout provides increased sensitivity over the optical density reading, since the background signal is decreased and issues such as cell clumping or inhomogeneities in the culture medium do not affect the output signal.

The dynamics of the effects on growth in this system are determined by the two players, MazEF and Mpro. At the onset of an experimental cycle, *MazEF* expression is kept low by the high methionine concentration in the preculture medium, while both Mpro and mCherry are expressed at high constitutive levels. On transfer to the culture medium, the external methionine concentration drops, and transcription of *MazEF* from the *MET3* promoter starts to rise. The toxic effect of active MazF is to degrade cellular RNAs. To minimize any complications from MazF cleaving its own mRNA, we engineered the coding region of *MazF* to lack ACA sequences. In the mRNA encoding Mpro, ACA sequences were left intact, however. By expressing Mpro from the strong

mSystems®

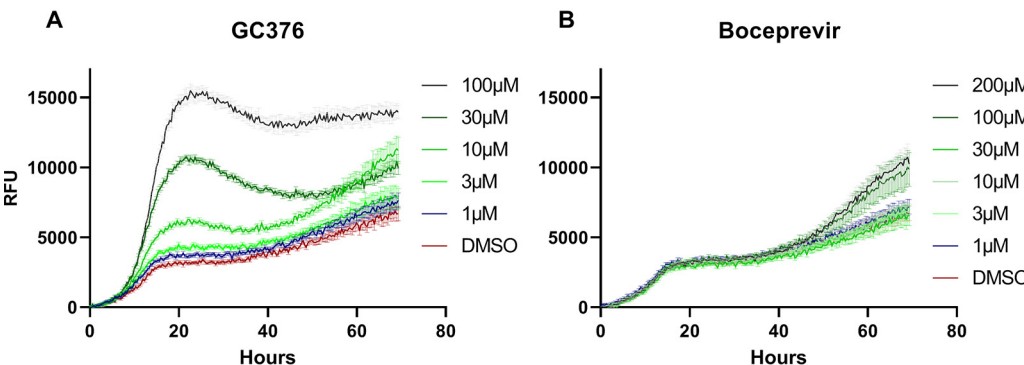

**FIG 5** Titration of GC376 and Boceprevir over wider concentration ranges. The same yeast strain, culture conditions, and growth measurements (fluorescence), and statistics were used described as for Fig. 4. GC376 was tested at 0, 1, 3, 10, 30, and 100 $\mu$M; boceprevir was tested at 0, 1, 3, 10, 30, 100, and 200 $\mu$M.

constitutive *Pichia GAP* promoter, we ensure that enough protease is present at the time MazEF production starts to increase through the gradual reduction of methionine in the medium from the experiment onset. Beside its intended target in the MazEF synthetic linker, Mpro can cleave endogenous yeast proteins. The consensus protease cleavage site profile of SARS-CoV-2 Mpro comprises the sequence [A/S/T]LQ[A/S/G] (4, 27), and there are 136 potential Mpro cleavage sites in the yeast proteome (see Table S2). There is a slight negative impact of expressing Mpro alone in yeast cells (see Fig. S5), as expected from cleavage of the cellular proteins. However, the expression of MazEF had no effect on growth in the absence of Mpro (see Fig. S2 and S3). This indicates that there is no background cleavage of MazEF by yeast proteases.

We also observed that starting the experiment at a low culture density (OD$_{600}$ < 0.1), such as presented here, works well for detecting Mpro inhibitors. A higher density of the preculture results in reduced sensitivity to the inhibitor, presumably because not enough MazF accumulates before the culture approaches stationary phase for it to significantly affect growth.

In the present system, GC376 gave a strong and specific signal as an Mpro inhibitor. Other molecules previously reported by various criteria to be Mpro inhibitors did not score, in the case of boceprevir not even at 200 $\mu$M. There are several possible explanations for this. First, the previous evidence from, e.g., an *in vitro* assay, could be misleading, as exemplified by aggregation-prone TDZD-8 (20). Another reason why a small molecule inhibitor may fail to show as a hit in this assay is insufficient cellular uptake. Yeast cells have multiple membrane-bound proteins, notably ABC transporters, which extrude exogenous molecules from the cell. Several attempts have been made to diminish this effect of transporters in order to sensitize yeast to small molecules, a key issue for chemical genomics and drug screening in yeast (28). A strain lacking all 16 ABC transporters has been generated (29), and a nine-tuple deletion strain lacking genes encoding transporters and transcription factors controlling similar genes displayed increased sensitivity to various compounds from ca. 2- to 200-fold (15). However, this strain has poor vigor, and it was later found that most of its sensitivity can be recovered by retaining only the *pdr1Δ pdr3Δ snq2Δ* gene deletions, which leaves the strain with robust growth properties (16). Here, we use this triple-deletion strain background, which should provide a higher sensitivity to small molecules and a higher scoring rate than in previous screens using the same physical cultivation and reading system and the same fluorescent marker, where only the *pdr5Δ* deletion was used to sensitize the yeast tester strain (23, 30, 31). Nevertheless, many compounds will not penetrate into the yeast cells and fail to score for that reason. Finally, inhibitors with general toxicity will not score in this assay system since the depression of growth will cancel out the positive signal from decreased protease activity. On the other hand, general toxicity is obviously to be avoided in this context. The

negative impact by TDZD-8 (Fig. 4F) could be due to general toxicity or to the compound inhibiting the yeast homologs of GSK3-$\beta$: Rim11, Mck1, Mrk1, and Ygk3 (32).

The setup is suitable for settings with robotic handling of samples at microtiter plate scale. The ability to continuously read the fluorescence and optical density of cultures during an entire growth cycle, with the ability to oxygenate the culture wells by shaking, is important to obtain reliable results. In a longer perspective, this setup may be modified to target other proteases by exchanging the protease cleavage site in the linker of the fusion protein. The concept is general but has some constraints. First, the consensus cleavage site of the protease has to be known with sufficient accuracy. Second, the protease has to be capable of acting in an environment where the fusion protein is accessible to attack, e.g., not in the interior channel of a proteasome.

In summary, we describe a functional *in vivo* screening system capable of identifying candidate inhibitors of SARS-CoV-2 Mpro. Being a cellular assay, it selects for molecules with high bioavailability while avoiding some pitfalls of *in vitro* environments. At the same time, it is performed in a safe laboratory setting, without virus particles. It is versatile in that it can be adapted to target other proteases, benefitting from the facile genetic engineering of *S. cerevisiae*. The system has been tuned to detect an inhibitor down to 1 $\mu$M using a positive selection mode. Importantly, this avoids the false positives in a negative selection mode, where compounds with general toxicity would score as hits.

## MATERIALS AND METHODS

**Construction of plasmids and yeast strains.** Details of the construction of each strain and plasmid are presented in Text S1, along with the full plasmid sequences. Primer sequences are given in Table S1.

**Candidate protease inhibitor compounds.** Boceprevir, GC376, TDZD-8, lercanidipine hydrochloride, efonidipine hydrochloride monoethanolate, and glycyrrhizic acid ammonium salt from *Glycyrrhiza* root were purchased from Sigma. Stocks of all the compounds were made in dimethyl sulfoxide (DMSO) at a concentration of 20 mM and stored frozen at −80°C.

**Culture conditions.** *S. cerevisiae* strains and plasmids used are listed in Table 1. YPD (2% glucose, 2% peptone, and 1% yeast extract) was used for routine culturing of strains not carrying plasmids. Synthetic defined (SD) medium (0.19% yeast nitrogen base, 0.5% ammonium sulfate, 2% glucose, and 0.077% Complete Supplement Mixture [CSM; ForMedium]) with appropriate dropout was used for routine culturing of strains carrying plasmids and for Bioscreen C (Labsystems Oy) phenotyping (uracil dropout with various concentrations of methionine). Liquid cultures were grown in a rotary shaker at 30°C at 200 rpm.

**Phenotypic analysis in Bioscreen.** Unless stated otherwise, overnight cultures were made in SD lacking uracil (SD–ura) containing 350 $\mu$M methionine and maintained at exponential growth phase. Before the start of experiment, the medium was removed, and the pellet resuspended in SD–ura–met to a final $OD_{600}$ of 1.0. Subsequently the cultures were diluted to an $OD_{600}$ of 0.1 into 100-well honeycomb plates (Labsystems Oy) prealiquoted with the appropriate media using an Opentrons OT2 robot to carry out both the dilution and the medium dispensing. Strains were cultivated for 3 days with the low shaking setting at 30°C with 20-min measurement intervals using a wide band filter (450 to 600 nm).

**Phenotypic analysis in robot Eve.** Overnight cultures were grown and maintained as described above. Before the start of experiment, medium was removed, and the pellet was resuspended in SD–ura containing 15 $\mu$M methionine to an $OD_{600}$ of 0.1. Within the automated workstation, the culture was aliquoted into a Greiner 384-well black plate with a clear bottom using a Thermo Combi Multidrop, and chemical compounds were transferred to the assay plate using the Bravo Liquid Handling platform to a final concentration of 10 or 30 $\mu$M for each compound (final DMSO concentration, 1.25%) and a final volume of 80 $\mu$l. Cells were incubated, and their growth was monitored essentially as described previously (23). Growth was stationary at 30°C, except at each 20-min interval, and samples were circularly agitated at 1,000 rpm for 10 s, followed by 10 s in the opposite direction. A read-and-incubate cycle was performed using the Overlord automation system to determine growth at 20-min intervals for 64 h. Fluorescence (580-nm excitation/612-nm emission) and $OD_{600}$ measurements were obtained with a BMG Polarstar plate reader.

## SUPPLEMENTAL MATERIAL

Supplemental material is available online only.

**TEXT S1**, DOCX file, 0.03 MB.

**FIG S1**, PDF file, 0.2 MB.

**FIG S2**, PDF file, 1 MB.
**FIG S3**, PDF file, 0.8 MB.
**FIG S4**, PDF file, 0.4 MB.
**FIG S5**, PDF file, 0.2 MB.
**TABLE S1**, DOCX file, 0.01 MB.
**TABLE S2**, XLSX file, 0.02 MB.

## ACKNOWLEDGMENT

This study was supported by a grant from the Swedish Research Council (2020-05738).

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
