## [Reviewer comments · mSystems]

A genetic trap in yeast for inhibitors of SARS-CoV-2 main protease

Hanna Alalam, Sunniva Sigurdardóttir, Catarina Bourgard, Ievgeniia Tiukova, Ross King, Morten Grötli, and Per Sunnerhagen

Corresponding Author(s): Per Sunnerhagen, University of Gothenburg

Review Timeline:

Submission Date:	September 12, 2021
Editorial Decision:	October 5, 2021
Revision Received:	October 22, 2021
Accepted:	November 3, 2021

Editor: Michelle Heck

Reviewer(s): The reviewers have opted to remain anonymous.

Transaction Report:

DOI: <https://doi.org/10.1128/mSystems.01087-21>

October 5, 2021

Prof. Per Sunnerhagen
University of Gothenburg
Dept. of Chemistry & Molecular Biology
P.O. Box 462
Lundberg Laboratory
Goteborg S-405 30
Sweden

Re: mSystems01087-21 (A genetic trap in yeast for inhibitors of SARS-CoV-2 main protease)

Dear Prof. Per Sunnerhagen:

Thank you for submitting your manuscript to mSystems. We have completed our review and I am pleased to inform you that, in principle, we expect to accept it for publication in mSystems. However, acceptance will not be final until you have adequately addressed the reviewer comments. I am looking forward to reading an edited draft.

Preparing Revision Guidelines

Sincerely,

Michelle Heck

Editor, mSystems

Journals Department
Reviewer comments:

Reviewer #1 (Comments for the Author):

Review of Alalam et al. "A genetic trap in yeast for inhibitors of SARS-CoV-2 main Protease"

In this study Alalam et al. describe the development of a yeast screening platform to identify small molecules that are capable to inhibit Mpro, the main protease of Sars-CoV-2. The screen uses a yeast strain expressing a toxin anti-toxin fusion protein (MazF toxin, MazE antitoxin) with an included cleavage site for Mpro between the two components of the fusion protein. Expression of Mpro in this yeast strain results in cleavage of the fusion protein and thus a MazF toxin mediated growth defect can be observed. Expression of mCherry as a marker protein in the yeast cells allows the detection of yeast growth using the fluorescence signal of the culture, which also makes the platform suitable for screening of larger compound libraries. The small molecule GC376, which can effectively inhibit Mpro, is used to demonstrate the functionality of this screening platform.

Points for consideration

*) The rescue of the growth rate can only be demonstrated for one compound. All the others show no effect or even increased toxicity (TDZT8). This raises some questions with regards to how to the general usefulness of the method. However, the results are well represented and illustrated and discussed.

*) Page2, paragraph "Importance": "The COVID-19 pandemic may continue several years before vaccination campaigns can put an"

First sentence a word is missing.

*) Fig.1: Color: red yeast with green border is hard to look at, maybe pick a smoother combination.

*) Fig.4: Labels for y-axes are missing.

Reviewer #2 (Comments for the Author):

In this work, Alalam et al report the development of a yeast-based experimental platform for screening small molecule inhibitors of the SARS-CoV-2 MPro protease. The authors present an interesting system, in which the MPro protease cleaves a linker peptide connecting a fusion between MazE and MazF (toxin/antitoxin E. coli system), which allows for the positive selection of yeast cells exposed to protease inhibitors. The system allowed the detection of GC376, an established Mpro inhibitor. Other five compounds previously reported as MPro inhibitors in vitro and/or in vivo assays were tested, but did not score in the present assay. Overall, the data presented is easy to follow and the strategy developed is interesting and has potential to increase the throughput of screenings for inhibitors of MPro and future variants. However, my enthusiasm was dampened by the fact that while the results demonstrate that the system has the capability to detect a protease inhibitor, its efficiency to detect true positives is less convincing.

Main comments:

1.Considering the main goal of the study is to establish an experimental platform for screening small molecule inhibitors of MPro protease, I consider fundamental that the results demonstrate 1) that the screen is able to positively score inhibitory compounds, which the authors did well using GC376, and 2) some indication of the efficiency with which the assay will correctly score true positives/negatives. Five compounds previously reported to inhibit MPro in in vitro and/or in vivo assays (including using human cells) did not score in the platform developed, which seems to challenge the usability of the assay as a screening method. The authors discuss this could be due to misleading evidence from previous assays, insufficient cellular uptake or general toxicity. These all seem plausible explanations, but the question of how frequently true positives are missed is still left unanswered. Is it possible to provide experimental evidence that the compounds tested in fact entered the cells? Given the systems allows for large-scale screens, a higher number of molecules could also be tested to provide more insight into these aspects.

2.It is unclear to me why the 10-30 uM concentration was chosen for all compounds in Fig. 4. It seems possible that some compounds might not have appeared as hits because their concentration was not sufficient to inhibit MPro. The authors may wish to perform tests exposing cells to higher concentrations of each compound to evaluate this possibility.

3.Figures 2 and 3 seem to demonstrate that there is a sweet spot concentration of methionine and GC376 that provides best separation between cells exposed or not to the inhibitor. Is it possible that methionine and inhibitor concentrations would have to be tuned for each compound of interest?

4.Minor: typo in page 4 "two protein moieties permi release"

Reviewer #1 (Comments for the Author):

Review of Alalam et al. "A genetic trap in yeast for inhibitors of SARS-CoV-2 main Protease"

In this study Alalam et al. describe the development of a yeast screening platform to identify small molecules that are capable to inhibit Mpro, the main protease of Sars-CoV-2. The screen uses a yeast strain expressing a toxin anti-toxin fusion protein (MazF toxin, MazE antitoxin) with an included cleavage site for Mpro between the two components of the fusion protein. Expression of Mpro in this yeast strain results in cleavage of the fusion protein and thus a MazF toxin mediated growth defect can be observed. Expression of mCherry as a marker protein in the yeast cells allows the detection of yeast growth using the fluorescence signal of the culture, which also makes the platform suitable for screening of larger compound libraries.

The small molecule GC376, which can effectively inhibit Mpro, is used to demonstrate the functionality of this screening platform.

Points for consideration

*) The rescue of the growth rate can only be demonstrated for one compound. All the others show no effect or even increased toxicity (TDZT8). This raises some questions with regards to how to the general usefulness of the method. However, the results are well represented and illustrated and discussed.

- GC376 has been consistently shown to be effective in targeting the major protease and used as a positive control in multiple studies. For the other drugs it is important to point out that even if the studies were done *in vivo*, usually a single cell line was used for screening purpose. It has now been shown that SARS-CoV-2 replication varies between cell lines (Wurtz *et al.* Eur J Clin Microbiol Infect Dis 40:477–484, 2021, Bakowski *et al.* Nature Commun 12:3309, 2021). Additionally, viral replication assays cannot accurately determine whether the observed effect of a compound is due to Mpro inhibition or something else, as there can be off-targets. Hence, a molecule that scored as a hit in such an assay may well not score in our type of screen, which target-based.

*) Page2, paragraph "Importance": "The COVID-19 pandemic may continue several years before vaccination campaigns can put an"

First sentence a word is missing.

- Corrected (actually two words missing)

*) Fig.1: Color: red yeast with green border is hard to look at, maybe pick a smoother combination.

- A mistake on our part. We have changed the border to dark red.

*) Fig.4: Labels for y-axes are missing.

- The labels were put on top of the diagrams to improve readability, rather than the conventional vertical text to the left of the Y-axes

Reviewer #2 (Comments for the Author):

In this work, Alalam et al report the development of a yeast-based experimental platform for screening small molecule inhibitors of the SARS-CoV-2 MPro protease. The authors present an interesting system, in which the MPro protease cleaves a linker peptide connecting a fusion between MazE and MazF (toxin/antitoxin *E. coli* system), which allows for the positive selection of yeast cells exposed to protease inhibitors. The system allowed the detection of GC376, an established Mpro inhibitor. Other five compounds previously reported as MPro inhibitors in vitro and/or in vivo assays were tested, but did not score in the present assay. Overall, the data presented is easy to follow and the strategy developed is interesting and has potential to increase the throughput of screenings for inhibitors of MPro and future variants. However, my enthusiasm was dampened by the fact that while the results demonstrate that the system has the capability to detect a protease inhibitor, its efficiency to detect true positives is less convincing.

Main comments:

1. Considering the main goal of the study is to establish an experimental platform for screening small molecule inhibitors of MPro protease, I consider fundamental that the results demonstrate 1) that the screen is able to positively score inhibitory compounds, which the authors did well using GC376, and 2) some indication of the efficiency with which the assay will correctly score true positives/negatives. Five compounds previously reported to inhibit MPro in in vitro and/or in vivo assays (including using human cells) did not score in the platform developed, which seems to challenge the usability of the assay as a screening method. The authors discuss this could be due to misleading evidence from previous assays, insufficient cellular uptake or general toxicity. These all seem plausible explanations, but the question of how frequently true positives are missed is still left unanswered.

Is it possible to provide experimental evidence that the compounds tested in fact entered the cells? Given the systems allows for large-scale screens, a higher number of molecules could also be tested to provide more insight into these aspects.

- It is inevitable that the cellular uptake of small molecules differs widely between cells and species. This is true also of yeast-based assays, as has been shown in many studies involving screens of compounds in *S. cerevisiae* (e.g. Smith *et al.* *Pharmacol Therapeut* 127:156, 2010; Mor *et al.* *mBio* 6:e00647, 2015; Petrovic *et al.* *Microbiol Res* 199:10, 2017). So to some degree the lack of signal for a compound will be due to inefficient uptake, however this is true also for mammalian cell lines which may vary considerably between them (see also our comment to Reviewer #1, first item). Other potential reasons include influences of *in vitro* assays of pH or redox status. For cellular assays of viral replication or cell survival, the

possibility remains that the observed effect was mediated through another target than mPro. Our assay, by contrast, is both carried out in living cells (avoiding artifacts from inappropriate *in vitro* environment) and target-based (avoiding effects mediated through off-targets). The relative contributions of each factor will be challenging to determine even when considering a larger scale screen. These issues are considered in Discussion.

2. It is unclear to me why the 10-30 μM concentration was chosen for all compounds in Fig. 4. It seems possible that some compounds might not have appeared as hits because their concentration was not sufficient to inhibit MPro. The authors may wish to perform tests exposing cells to higher concentrations of each compound to evaluate this possibility.

- We intentionally used lower concentration of drugs to demonstrate the sensitivity of our system compared to a system that depended solely on the growth depression phenotype induced by SARS-CoV-1 protease in yeast (Frieman *et al.* PLoS One 6:e28479, 2011) without the amplified positive phenotypic selection made possible in our system by release of the MazF toxin. In their case 50 μM of each compound was used for the primary screen. Having a sensitive system able to detect a phenotypic difference at lower concentration is a clear advantage. As an example of the opposite, glycyrrhizin requires very high concentration to have an effect (1.2 mM; van de Sand *et al.* Viruses 13:609, 2021), putting its potential as an effective therapeutic in question.

- To meet this request from the reviewer for tests with a wider concentration range, we have now performed titrations from 1 to 100 μM of the protease inhibitor GC376, showing that we can detect its activity down to 1 μM . We also show the corresponding data for boceprevir, which did not display activity up to 30 μM . When tried at 100 - 200 μM , we only observe a very late growth stimulatory effect, indicating that at least for this substance, testing at higher concentrations does not improve sensitivity. These data are shown in the new Figure 5.

3. Figures 2 and 3 seem to demonstrate that there is a sweet spot concentration of methionine and GC376 that provides best separation between cells exposed or not to the inhibitor. Is it possible that methionine and inhibitor concentrations would have to be tuned for each compound of interest? As demonstrated in the figure, increasing the methionine concentration serves to decrease the sensitivity of the system as with higher methionine allows better growth. It is more desirable to adjust the compound concentration rather than the methionine concentration, however, excessively high concentration would likely lead to higher fraction of false positive. In that regard we prefer to apply a methionine concentration that highly sensitizes the system in combination with lower compound concentrations to allow robust separation of positive hits that have high confidence. As for attuning the compound/methionine concentration for each individual compounds, customarily this is done post screen since adjusting these conditions for each individual molecule for which the vast majority will test negative defeats the purpose of a high-throughput screen.

- As demonstrated in the figure, increasing the methionine concentration decreases the sensitivity of the system as with higher methionine allows better growth. We prefer to apply a methionine concentration that highly sensitizes the system in combination with lower compound concentrations to allow robust separation of positive hits with a high confidence.

As for tuning the compound/methionine concentration for each individual compounds, customarily this is done post screen since adjusting these conditions for each individual molecule out of thousands for which the vast majority will test negative defeats the purpose of a high-throughput screen.

4.Minor: typo in page 4 "two protein moieties permi release"

- This has been corrected ("permit")

November 3, 2021

Prof. Per Sunnerhagen
University of Gothenburg
Dept. of Chemistry & Molecular Biology
P.O. Box 462
Lundberg Laboratory
Goteborg S-405 30
Sweden

Re: mSystems01087-21R1 (A genetic trap in yeast for inhibitors of SARS-CoV-2 main protease)

Dear Prof. Per Sunnerhagen:

Your manuscript has been accepted, and I am forwarding it to the ASM Journals Department for publication. For your reference, ASM Journals' address is given below. Before it can be scheduled for publication, your manuscript will be checked by the mSystems senior production editor, Ellie Ghatineh, to make sure that all elements meet the technical requirements for publication. She will contact you if anything needs to be revised before copyediting and production can begin. Otherwise, you will be notified when your proofs are ready to be viewed.

As an open-access publication, mSystems receives no financial support from paid subscriptions and depends on authors' prompt payment of publication fees as soon as their articles are accepted. =

Publication Fees:

We recognize that the video files can become quite large, and so to avoid quality loss ASM suggests sending the video file via <https://www.wetransfer.com/>. When you have a final version of the video and the still ready to share, please send it to Ellie Ghatineh at eghatineh@asmusa.org.

Sincerely,

Michelle Heck
Editor, mSystems

Journals Department
File S1: Accept
Fig. S1: Accept
Table S1: Accept
Fig. S4: Accept
Table S2: Accept
Fig. S2: Accept
Fig. S5: Accept
Fig. S3: Accept